# COVID-19 and Venous Thromboembolism: From Pathological Mechanisms to Clinical Management

**DOI:** 10.3390/jpm11121328

**Published:** 2021-12-08

**Authors:** Xianghui Zhou, Zhipeng Cheng, Yu Hu

**Affiliations:** 1Department of Hematology, Union Hospital, Tongji Medical College, Huazhong University of Science and Technology, Wuhan 430022, China; dr_zxh@hust.edu.cn (X.Z.); dr_cheng@hust.edu.cn (Z.C.); 2Collaborative Innovation Center of Hematology, Huazhong University of Science and Technology, Wuhan 430022, China; 3Hubei Clinical Medical Center of Cell Therapy for Neoplastic Disease, Wuhan 430022, China

**Keywords:** COVID-19, VTE, pathophysiology

## Abstract

Coronavirus disease 2019 (COVID-19), which is becoming a global pandemic, is caused by SARS-CoV-2 infection. In COVID-19, thrombotic events occur frequently, mainly venous thromboembolism (VTE), which is closely related to disease severity and clinical prognosis. Compared with historical controls, the occurrence of VTE in hospitalized and critical COVID-19 patients is incredibly high. However, the pathophysiology of thrombosis and the best strategies for thrombosis prevention in COVID-19 remain unclear, thus needing further exploration. Virchow’s triad elements have been proposed as important risk factors for thrombotic diseases. Therefore, the three factors outlined by Virchow can also be applied to the formation of venous thrombosis in the COVID-19 setting. A thorough understanding of the complex interactions in these processes is important in the search for effective treatments for COVID-19. In this work, we focus on the pathological mechanisms of VTE in COVID-19 from the aspects of endothelial dysfunction, hypercoagulability, abnormal blood flow. We also discuss the treatment of VTE as well as the ongoing clinical trials of heparin anticoagulant therapy. In addition, according to the pathophysiological mechanism of COVID-19-associated thrombosis, we extended the range of antithrombotic drugs including antiplatelet drugs, antifibrinolytic drugs, and anti-inflammatory drugs, hoping to find effective drug therapy and improve the prognosis of VTE in COVID-19 patients.

## 1. Introduction

Until August 23, 2021, COVID-19 has gradually spread to multiple countries and regions as the epidemic worsens, with statistics showing about 211 million laboratory-confirmed cases and approximately four million deaths due to the progression of the disease. (https://www.who.int (accessed 23 August 2021)), leading to a profound impact on society, culture, and the global economy. COVID-19 patients may present with variable clinical features, ranging from asymptomatic infection to life-threatening clinical comorbidities [1]. Interestingly, apart from acute respiratory distress syndrome (ARDS), shock, and heart failure, liver failure, kidney failure, systemic coagulation derangement is increasingly recognized as an important clinical course in COVID-19 patients with upregulated coagulation indexes including fibrin/fibrinogen degradation products and D-Dimer [2,3]. In this regard, systemic activation of blood coagulation can cause pathological thrombotic events including arterial thrombosis, venous thrombosis, and microvascular thrombosis, indicating a poor prognosis such as DIC, pulmonary embolism, and other fatal manifestations. Furthermore, autopsy studies have identified pulmonary embolism, diffuse alveolar damage, and a high occurrence of deep venous thrombosis (DVT) (58%). In this study, imaging examination of deceased patients revealed reticular infiltration in the lungs with severe bilateral compactness and consolidation. About 67 percent of the patients had diffuse lung injury. In all patients, SARS-CoV-2 RNA was detected in multiple organs with high concentrations, mainly in the lungs [4]. As the epidemic progresses, accumulated studies have revealed an increased incidence of VTE in patients hospitalized with SARS-CoV-2, especially if they become critical illness with COVID-19 [5]. There is a comprehensive systematic review from multiple health care systems, showing that the occurrence of VTE in COVID-19 inpatients was 17.3%, of which approximately 2/3 were DVTs. According to the subgroup analysis, VTE occurrence was more common in the imaging diagnostic group (33.1% vs. 9.8% with clinical diagnosis), and intensive care unit group (27.9% vs. 7.1% in the general ward) [6]. Importantly, the incidence of VTE was also associated with mortality in patients with COVID-19, as studies indicated that significant mortality is thought to be secondary to VTE. In fact, autopsy results of COVID-19 deaths suggest that thrombosis plays an important role in mortality [3,4,7]. According to previous studies, the risk factors for VTE are diverse including immobilization, malignant tumors, major surgery, a history of VTE, and chronic heart disease [8]. Beyond that, age, gender, BMI, and lymphocyte levels may be closely related to the incidence of VTE. Inflammatory factors and coagulation factors (CRP, D-dimer, APTT, FDP) can be used to predict the occurrence of venous thrombosis [9]. Clinical analysis showed that critically ill patients with COVID-19 infection have a nearly 10-fold increased risk of venous thromboembolism or death [10].

However, the values of these factors are not equal in patients with COVID-19 infection. Therefore, on account of the implications in the clinical diagnosis, anticoagulant prophylaxis, and treatment, understanding the underlying pathophysiological mechanisms of VTE in the progression of COVID-19 is of great significance. In the subsequent discussion, we review the existing research on the involvement of VTE with SARS-CoV-2 infection, focusing on the latest evidence on the pathologic mechanisms and clinical management of VTE.

## 2. Pathophysiology of VTE in COVID-19

In exploring the pathophysiology of VTE formation, we should remember what the famous German physician Rudolph Ludwig Karl Virchow described as the three major factors contributing to thrombosis, namely: endothelial dysfunction, hypercoagulable state, and blood stasis (the Virchow triad) [11]. The infection with SARS-CoV-2 can enhance three aspects of the Virchow triad, resulting in an increased risk of thrombosis, which will be elaborated on these three characteristics below.

### Endothelial Dysfunction

Endothelial cells, as a single-cell layer on the inner wall of blood vessels, act as a mechanical barrier between blood and basement membrane (a protective layer of limiting endothelial cells (EC)–immune cell and EC–platelet interactions), controlling vascular tone and immunomodulation [12]. Vascular endothelium is a highly active paracrine, endocrine, and autocrine organ in the secretion system. Cytokines secreted by vascular endothelium play an important role in regulating angiotasis and maintaining vascular homeostasis [13]. Endothelial dysfunction is a major contributor to microvascular dysfunction including endothelial activation and decreased vasodilation, resulting in proinflammatory, hypercoagulable, and proliferative states [14,15]. Among the pathological mechanisms of VTE in COVID-19, endothelial dysfunction could be induced by many factors including direct SARS-CoV-2 invasion of ECs or secondary inflammation [16,17].

First, SARS-CoV-2 directly infects ECs and results in diffuse endothelial inflammation, dysfunction, apoptosis, and pyroptosis, thus impairing the normal antithrombotic activity of endothelial cells [4,18]. Second, to enter cells, emerging evidence demonstrates that SARS-CoV-2 targets multiple organs, expressing the angiotensin converting enzyme 2 receptor (ACE2) [19]. Existing research suggests that ACE2 receptor is widely expressed on endotheliocytes. In addition, there is a kind of serine protease (transmembrane protease serine 2) that can cleave and activate spike proteins on the membrane, promoting the membrane fusion of SARS-CoV-2. Then, endocytosis and proliferation of SARS-CoV-2 may continue, eventually leading to infection [17,20,21]. Moreover, SARS-CoV-2 virus binds ACE2 and intracellular translocation, impairing ACE2 and depriving it of endogenous normal enzyme function. Under these conditions, it is conceivable that depletion of ACE2, characteristic of SARS-CoV-2 infection, facilitating the kallikrein-bradykinin (BK) system, inflammation, vascular permeability, and coagulopathy [22,23]. Furthermore, immune cells and inflammatory cytokines could enhance endothelial cell contraction and lead to relaxation of endothelial junctions. This, in turn, pulls the endothelial cells apart, resulting in the opening of gaps between adjacent endothelial cells [24]. In addition, increased levels of proinflammatory cytokines (Interleukin-1, Interleukin-6, and TNF) were found in COVID-19 patients, which may explain the endothelial dysfunction to some degree. Previous works have demonstrated that endothelial cells can secrete a variety of anticoagulant, antithrombotic substances. However, these molecules are normally covered with glycoprotein, which could insulate them from blood cells such as red cells, platelets, and immune cells. When infected with SARS-CoV-2, the breakdown of glycocalyx would lead to the activation of endothelial cells, eventually resulting in thrombotic events [25,26]. Finally, when hypoxia occurs in COVID-19 patients, expressions of membrane adhesion molecules (P-selectin, E-selectin, ICAM-1, and VCAM-1) may activate the COX signaling pathway in ECs, resulting in endothelial cell destruction and vascular smooth muscle cell contraction [27]. Other studies have demonstrated that hypoxia may result in elevating hypoxia inducible factor-1α (HIF-1α), expressed by endothelial cells and immune cells, and more importantly, it could enhance endothelial cell damage in COVID-19 patients by reducing CD55 expression [28]. In conclusion, endothelial injury has great implications for the formation of VTE in COVID-19 patients, not only in terms of its structure and function, but also in the cascade reaction caused by it.

## 3. Hypercoagulable State

When SARS-CoV-2 infection occurs, The coagulation changes indicate a hypercoagulable state, and these changes may increase the risk of thromboembolic complications. The induction of a procoagulant state along with abnormal markers of coagulation (complement and cytokines), increased platelet reactivity, neutrophil extracellular traps (NETs). We will describe their pathophysiological mechanisms in the following discussion.

### 3.1. Platelet Activation

Platelets, a type of non-nuclear blood cell in circulating blood, are known for their important role in thrombosis and hemostasis [29]. Recent research has found that the occurrence of thrombotic complications is particularly high in critically ill patients with COVID-19. Increased arterial and venous thrombosis, even pulmonary embolism, are consistent with the fact that increased platelet activation is more likely to occur in severe patients than in mild or asymptomatic subjects [30]. All of the data placed platelets as conductors of deep venous thrombosis (DVT) pathogenesis, the mechanics of which have been explored. First, the SARS-CoV-2 virus can directly induce platelet activation through Spike/ACE2 interactions, further causing thrombosis. In this process, studies showed that mitogen-activated protein kinase (MAPK) was involved in SARS-CoV-2-induced platelet activation, which also explained the phenomenon that the MAPK signaling pathway of platelets was stimulated in COVID-19 patients [31,32]. On the other hand, viral infection could trigger a range of inflammatory responses, and immune substances that may also lead to high platelet activity in COVID-19 patients [31]. In COVID-19, some of the dysregulated pro-inflammatory cytokines (IL-1β, IL-6, and IL-8) play a pathological role in promoting platelet activation and causing platelet dysfunction. For example, IL-1β could regulate platelet aggregation through its receptor IL-1R1 expressed on platelets [33]. In addition, other mechanisms related to the pathogenesis of COVID-19 include hypoxemia, increased inflammation, immune system activation, platelet apoptosis, and endothelial cell dysfunction, [34,35], which might further promote platelet activation, ultimately leading to thrombosis [36]. Previous articles have demonstrated that SARS-CoV-2-induced pathological inflammation promotes the increased expression of P-selectin and the release of sCD40L, which together with thrombin promotes platelet activation. This, in turn, further increases thrombin, p-selectin, and sCD40L levels, creating a positive feedback loop that promotes platelet activation and thrombosis [33].

### 3.2. Cytokines and Chemokines

In the process of COVID-19, elevated plasma levels of IL-6, IFN-γ, IL-2, IL-7, IL-15, G-CSF, MCP1, MIP1α, and TNF were observed [25,37,38]. Multiple pro-inflammatory cytokines such as IFN-γ, IL-6, and IL-2 can induce the formation of hypercoagulable state, conducive to the occurrence of thrombotic events. Among them, IFN-γ mainly causes thrombosis by stimulating platelet activity and damaging vascular endothelium [39]. It can also promote the formation of venous thrombosis by mediating inflammatory response and NETs [40]. Studies have also found that the combination of TNFα and IFN-γ could facilitate the death of inflammatory cells through activating the JAK/STAT1/IRF1 axis during SARS-CoV-2 infection, which could be a trigger for inflammatory thrombosis [41]. IL-6 leads to hypercoagulability by promoting platelet activation, endothelial dysfunction, promoting coagulation factor disorders [42]. In the case of IL-2, it can induce cytokines to impair the anticoagulation of endothelial cells, leading to activation of the coagulation system [43]. In short, mechanistic understanding of pro-inflammatory cytokines including TNF-a, IL-1, or IL-6 might be a basis to support the utility of the treatment in inflammatory thrombosis.

### 3.3. Complement

Complement involves a cascade of processes to facilitate the expression of tissue factor (TF) and induce a pre-thrombotic phenotype, eventually implicated in the formation of thrombosis [44]. First, complement can enhance neutrophil/monocyte activation, and complement effectors such as platelets, which could promote thrombotic inflammation, microvascular thrombosis, and endothelial dysfunction [45]. Over the process of COVID-19, studies have found that specific IgM increases during the acute phase and specific IgG increases during subsequent phases, which can produce immune complexes that lead to inflammation, coagulation, and further activation of the complement system [46]. In addition, when infected with SARS-CoV-2, complement activation may be an important mediator to regulate a proinflammatory response. Complement activation pathways include the classical pathway, alternative pathway, and lectin pathway, which lead to the production of C3a and C5a. In turn, it stimulates the formation of the C5b–9 membrane attack complex (MAC) to recruit neutrophils, ultimately leading to endothelial inflammation and the induction of a prethrombotic state [47]. Therefore, understanding the association between the complement and prothrombotic state during COVID-19 is helpful for the treatment of thrombotic diseases including venous thrombosis.

### 3.4. Neutrophil Extracellular Traps/NETs

Neutrophil extracellular traps are three-dimensional lattices composed of densified chromatin coated with histone and antimicrobial proteins that promote immune thrombosis [48]. NETs are composed of chromatin fiber mesh, antimicrobial peptide particles, and enzymes released by neutrophils in order to control infections [49]. In recent works, neutrophil infiltration was detected in the pathologic results from the autopsy of COVID-19 deaths [50]. Moreover, increased peripheral blood neutrophil counts were found in patients with severe and non-surviving COVID-19 [51]. Previous research has indicated that NETs may be important markers of disease severity in COVID-19 patients. More importantly, it has the potential to be an important player in COVID-19 immune thrombosis [48,52]. Recent studies have shown that platelet-white blood cell interactions are important in promoting platelet and neutrophil activation [53]. One study detected increased levels of PF4 during the course of COVID-19 infection. PF4 is released from platelets and binds to NETs, making them resistant to DNA enzymes [54]. P-selectin (CD62P) mediates the connection between platelets and neutrophils by binding p-selectin glycoprotein ligand 1 (PSGL-1) on neutrophils to promote platelet activation [55]. Furthermore, cathepsin G and neutrophil elastase in NETs could regulate platelet function, and also activate platelets by protease-activated receptor 4 to promote fibrin formation. Their cleavage results in activation of glycoprotein IIb–IIIa as well as activation of plasma enzymes such as coagulation factors X and V, ultimately leading to complete platelet activation [56,57] Moreover, recent studies have demonstrated a connection between lysophosphatidic acid (LPA) and NETs. LPA is a bioactive phospholipid from activated platelets and itself could induce the production of NETs, further activating platelets to generate LPA. This positive feedback between LPA, NET, and platelets can lead to the disordered immune thrombotic state [58,59].

## 4. Abnormalities of Blood Flow

Among the risk factors of venous thrombosis, not only endothelial injury and the formation of hypercoagulability, but also abnormal blood flow are important factors of venous thrombosis. Blood stasis usually occurs as a result of prolonged bedrest, immobilization, strict isolation, and limited physiotherapy, especially in critically ill patients. During the course of SARS-CoV-2 infection, patients with fatigue, hypoxemia, connection to medical equipment (ventilator, electrocardiogram monitor), or serious complications (respiratory failure, heart failure, or other organ involvement) are more likely to develop venous thrombosis because they have limited movement [60]. In addition, during COVID-19 acute respiratory distress syndrome (ARDS), the disease can cause pulmonary artery resistance and increased right ventricular pressure, leading to pulmonary microcirculation thrombosis, followed by venous return to the heart with reduced blood flow, resulting in peripheral venous stasis [16,61].

## 5. VTE Clinical Management

The high occurrence of venous thrombus observed in patients with critically ill COVID-19 has aroused people’s interest in the methods of prophylaxis and treatment of venous thrombosis for patients with COVID-19. In this regard, according to the mechanism of venous thrombosis and the characteristics of Virchow’s triad, there are a series of possible prophylactic and therapeutic drugs including anticoagulant, antiplatelet drugs, and other related drugs, and are described in detail below. In addition, as of 28 August 2021, we found 20 studies of clinical trials from the ClinicalTrials.gov database, focusing on the interventions to prevent and treat VTE in COVID-19 patients. These trials are investigating different doses of heparin and low molecular weight heparin, different doses of platelet inhibitors, and other drugs, as listed in Table 1.

### 5.1. Anticoagulation Management

Low molecular weight heparin (LMWH) or unfractionated heparin (UFH) is recommended as a prophylactic antithrombotic therapy. This has been endorsed by the Scientific and Standardization Committee on Coagulation of the International Society for Thrombosis (SCC-ISTH). They have the advantage of having a short half-life, being easy to administer (IV or subcutaneous), and having fewer drug interactions than oral anticoagulants. If pharmacological prophylaxis is contraindicated, SCC-ISTH recommends that a mechanical approach should be considered for multimodal thromboprophylaxis [62]. Interestingly, the application of heparin has many benefits, not only an anticoagulant effect, but also a variety of other effects. Studies have demonstrated that heparin has some protective effects on endothelial cells in addition to its possible anti-inflammatory effects in COVID-19 patients, and may also be helpful in an anti-viral effect for coronavirus infection [63,64,65].

Of note, low molecular weight heparin (LMWH) is of great significance for post-discharge treatment, since patients are also at risk for venous thromboembolism after discharge, up to 90 days after hospitalization. Therefore, susceptibility to VTE also needs to be considered in the infection of SARS-CoV-2. SCC-ISTH recommends consideration of low-molecular-weight heparin or FDA-approved prophylactic anticoagulants after discharge (rivaroxaban and betrixaban) [65]. Although heparin therapy is a cornerstone in the prevention and treatment of COVID-19 patients., its additional effects should not be ignored. For example, heparin-induced thrombocytopenia (HIT) is a severe immune-mediated prothrombotic disorder to heparin as it could form complexes with platelet factor 4 (PF4). This binding could induce anti-PF4/heparin IgG antibodies, which may stimulate platelet activation, leading to a prothrombotic syndrome [66]. Therefore, real-time monitoring of anti-PF4/heparin antibodies, together with clinical symptoms suggestive for HIT, are very important for the selection and application of heparin. In addition, it is noteworthy that the majority of thrombotic events occurred in severe patients because critically ill patients have more risk factors for thrombosis. For example, a study in New York reported significantly higher benefits of anticoagulant therapy in severely ill patients such as those on mechanical ventilation, with a 33.6 percent reduction in mortality [67]. Therefore, special attention should be paid to the selection of anticoagulant therapy in severe patients.

Direct oral administration of anticoagulants (DOACs) are commonly used in COVID-19 patients who require anticoagulation. Nevertheless, oral anticoagulants are not routinely administered in hospitalized COVID-19 patients as DOACs might interact with many anti-inflammatory agents and some antivirals, affecting the efficacy of drugs such as tocilizumab, lopinavir, and ritonavir. In addition, based on contraindications for its use, obstructed excretion of DOACs in patients with renal damage may cause a risk of bleeding. Therefore, attention should be paid to the use of oral anticoagulants [68,69,70].

### 5.2. Antiplatelet Agents

Several antiplatelet agents including tirofiban, dipyridamole, and nafamostat have also received attention in the treatment of complications in COVID-19 patients. There are ongoing clinical trials to verify their antithrombotic efficacy. For tirofiban, a single center, case control, phase IIb study (NCT04368377) was conducted in the L. Sacco Hospital Milano, Lombardia, Italy. They found that antiplatelet agent (tirofiban) may effectively improve ventilation/perfusion ratio in COVID-19 patients with critical respiratory failure, possibly affecting the prevention of thrombosis, intervening in megakaryocyte function and platelet adhesion [71]. Regarding dipyridamole, recent work has reported that it was able to improve the clinical prognosis in severe COVID-19 [72]. An ongoing clinical trial (NCT04391179) conducted at the University of Michigan Ann Arbor, Michigan, United States. The aim of the study was to assess whether the application of dipyridamole lasting 14 days would reduce excessive coagulation of COVID-19. The results of this trial will be eagerly awaited. Finally, nafamostat has been used for pancreatitis, DIC, and dialysis in Japan for more than 30 years. Nafamostat is an inhibitor of the synthesis of serine proteases (thrombin, plasminase, and trypsin) with antiviral, anti-inflammatory, and anticoagulant activity, and its role is being evaluated [73]. Importantly, unlike heparin, namorestat does not cause bleeding side effects even at doses used for DIC. This major advantage is due to the strong anti-fibrinolytic effect of nafamostat [74]. In addition, nafamostat is known to inhibit TMPRSS-2, which shows a significant role during the course of COVID-19 [75]. Nafamostat promises to be a promising solution for COVID-19, which is characterized by hypercoagulability and enhanced fibrinolysis of venous thromboembolism (VTE).

### 5.3. Other Clinical Management

In COVID-19 patients, induction of apoptosis and pyrodeath may play an important role in endothelial cell damage, and it provides a theoretical basis for treatment to stabilize endothelial cells while coping with viral replication, especially in the application of anti-inflammatory, anti-cytokine drugs and ACE inhibitors [35]. Recently, endothelial cell stabilization has clearly been identified as a therapeutic target for COVID-19. Adrecizumab (HAM8101) is an anti-adrenomedullin (anti-ADM) antibody that targets vascular and capillary leakage in sepsis and inflammation and ultimately stabilizes and maintains endothelial barrier function [76]. In addition, cytokine release syndrome is one of the most alarming complications of COVID-19 virus infection, and cytokine-targeted agents may confirm their role in the treatment. Among them, tocilizumab, a monoclonal antibody against the interleukin-6 (IL-6) receptor, has been found to reduce mortality and hospitalizations to the ICU, and the degree of systemic inflammation. More interestingly, it has been shown that tofacitinib could block the pathways of several important anti-inflammatory cytokines (IL 2, 4, 7, 9, 15, and 21), while it does not inhibit the IL10 signaling pathway (a main anti-inflammatory factor) [77,78]. These characteristics may explain why the use of tofacitinib did not develop with significant side effects [79]. In addition, studies suggest that inhibiting the high activity of platelets through the application of recombinant human ACE-2 protein and anti-Spike monoclonal antibodies may be an effective treatment for inhibiting VTE in patients with COVID-19 [31].

In COVID-19, given the direct role of NETs in immune thrombosis, blocking NETs may be helpful to improve patient outcomes. With current research advances, the new developed NETosis inhibitors including Lonodelestat, Alvelestat, CHF6333, and Elafin [80] may be therapeutic options for COVID-19 thrombotic.

Based on available clinical data of COVID-19, fibrinolysis shutdown was observed in severe patients, and increased occurrence of VTE has been seen in patients with severe thrombolysis abnormalities [81]. Therefore, fibrinolytic agents are being assessing due to their potential therapeutic function in COVID-19 such as tissue-type plasminogen activator [82].

Finally, evidence is accumulating on complement-related thrombotic microangiopathy in COVID-19 patients [83]. Currently, two FDA-approved complement inhibitors (eculizumab, ravulizuma) that can combine with C5 and inhibit cleavage with C5a and C5b suppress the production of MAC [84,85].

## 6. Conclusions

In COVID-19, there are a variety of factors affecting the occurrence and development of VTE, but the risk factors involved can be more systematically clarified from the perspective of Virchow triad. On one hand, endothelial dysfunction, hypercoagulable state, and abnormal blood flow are respectively responsible for the extremely high incidence of VTE events in patients with COVID-19, while on the other hand, they interact with each other. For example, a feedback pathway promotes the occurrence of inflammation, which leads to the high activity of platelets and ultimately causes the occurrence and maintenance of venous thrombosis. It can be seen from our discussion that the prevention and treatment of SCC-ISTH, LMWH, and UFH are the first choice, but the treatment and management of antithrombotic therapy have multiple targets according to the pathophysiological mechanism of thrombosis. Therapies targeting these pathologic mechanisms may alleviate venous thrombosis in COVID-19 including anticoagulants, antiplatelets, fibrinolytic agents, and immunomodulators. This provides a theoretical basis for heparin replacement therapy and combination therapy, but the choice of specific regimen needs to be verified by future clinical trials. In summary, understanding the complex mechanisms between COVID-19 virus infection, vasculature, immune system, and coagulation can contribute to effective treatment for clinical management.

## Figures and Tables

**Table 1 jpm-11-01328-t001:** Ongoing trials focusing on interventions for the prevention and treatment of VTE in COVID-19 patients up to 28 August 2021.

ClinicalTrials.gov Identifier	Study Design/Status	Patients, No.	Treatment Group
NCT04842292	Interventional/Recruiting	40	Heparin vs. Placebo
NCT04746339	Interventional/Recruiting	1000	Apixaban 2.5 mg vs. Placebo
NCT04650087	Interventional/Recruiting	5320	Apixaban 2.5 mg vs. Placebo
NCT04600141	Interventional/Recruiting	308	Tocilizumab vs. Heparin Therapeutic dosage vs. Heparin Prophylactic dosage
NCT04581954	Interventional/Recruiting	456	Ruxolitinib vs. Fostamatinib vs. Standard of care
NCT04542408	Interventional/Recruiting	172	Anticoagulation Agents (Edoxaban and/or high dose LMWH) vs. Low dose Low molecular weight heparin or Placebo
NCT04508023	Interventional/Recruiting	4000	Rivaroxaban vs. Placebo vs. Standard of Care (SOC)
NCT04492254	Interventional/Recruiting	1370	Enoxaparin
NCT04486508	Interventional/completed	600	Intermediate dose Enoxaparin/unfractionated heparin vs. Standard prophylactic dose Enoxaparin/unfractionated heparin vs. Atorvastatin 20 mg vs. Matched placebo
NCT04466670	Interventional/Recruiting	379	Acetylsalicylic acid vs. Unfractionated heparin nebulized
NCT04416048	Interventional/Recruiting	400	Rivaroxaban vs. Standard Of Care (SOC)
NCT04406389	Interventional/Recruiting	186	Enoxaparin sodium vs. Unfractionated heparin vs. Fondapariniux vs. Argatroban
NCT04401293	Interventional/completed	257	Enoxaparin vs. Prophylactic/Intermediate Dose Enoxaparin
NCT04394377	Interventional/completed	615	Rivaroxaban20 mg/d followed by enoxaparin/unfractionated heparin when needed vs. Control group with enoxaparin 40mg/d
NCT04394000	Interventional/completed	72	Thromboprofylaxis protocol vs. Standard protocol
NCT04373707	Interventional/Recruiting	602	Enoxaparin
NCT04372589	Interventional/completed	1200	Heparin
NCT04367831	Interventional/Recruiting	100	Enoxaparin Prophylactic Dose vs. Heparin Infusion vs. Heparin SC vs. Enoxaparin/Lovenox Intermediate Dose
NCT04366960	Interventional/completed	189	Enoxaparin
NCT04360824	Interventional/Recruiting	170	Intermediate dose thromboprophylaxis vs. Standard of Care thromboprophylaxis

## Data Availability

Graphical Abstract is created with BioRender.com (accessed on 7 December 2021).

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
