# Peer review of "COVID-19 and Venous Thromboembolism: From Pathological Mechanisms to Clinical Management"

_jpm, 2021, doi:10.3390/jpm11121328_

Round 1
Reviewer 1 Report
The review from Zhou et al. analyzed the main pathlogical pathways involved in COVID-19 venous thromboembolism and its clininal implications.
It is an important, well conceptualized review. Since the mechanisms and the pathways of thrombosis by Covid-19 are very complex, it would be important to provide a grafical abstract. The mistakes should be corrected.
Author Response
Thank you for your letter and for the reviewers’ comments concerning our manuscript entitled “COVID-19 and venous thromboembolism: from pathological mechanisms to clinical management” (jpm-1489876). Your comments are all valuable and very helpful for revising and improving our paper. We have studied comments carefully and have made correction which we hope meet with approval. The major revisions are underlined in the paper. The main corrections in the paper and the responds to the reviewer’s comments are as flowing:
Responds to the reviewer’s comments:
Reviewer #1:
Comment 1.1: It is an important, well conceptualized review. Since the mechanisms and the pathways of thrombosis by Covid-19 are very complex, it would be important to provide a graphical abstract. The mistakes should be corrected.
Response: Thank you very much for your comments and suggestions. In response to your questions, we have re-uploaded graphical abstract and attached it to the manuscript. Thanks again for your suggestions.
We tried our best to improve the manuscript. These changes will not influence the content and framework of the paper.
We appreciate for Editors/Reviewers’ warm work earnestly, and hope that the correction will meet with approval.
Once again, thank you very much for your comments and suggestions.
Reviewer 2 Report
Authors reviewed “COVID-19 and venous thromboembolism: from pathological mechanisms to clinical management”
- Authors should state the HIT antibody.
- A relationship between platelet activation and VTE is still not clear.
- Anticoagulant therapy was reported to be useful in severe cases but to be not useful in mild cases.
- When did VTE occur in patients with COVID-19?
- Which vessels were frequently obstructed by thrombosis?
- Illustration for the mechanism for onset of VTE in patients with COVID-19 may be helpful.
- Relationship mortality and VTE may be important.
- Risk factors for VTE in patients with COVID-19 may be helpful.
- "Take home message" may be required.
Author Response
Thank you for your letter and for the reviewers’ comments concerning our manuscript entitled “COVID-19 and venous thromboembolism: from pathological mechanisms to clinical management” (jpm-1489876). Your comments are all valuable and very helpful for revising and improving our paper. We have studied comments carefully and have made correction which we hope meet with approval. The major revisions are underlined in the paper. The main corrections in the paper and the responds to the reviewer’s comments are as flowing:
Responds to the reviewer’s comments:
Reviewer #2
Comment 2.1: Authors should state the HIT antibody
Response: Considering the Reviewer’s suggestion, we have supplemented the relevant literatures in the revised manuscript. In the VTE Clinical Management (1. Anticoagulation Management) section, we discussed the occurrence of heparin-induced thrombocytopenia, as well as the formation of its HIT antibodies. (lines 262-269) Thank you very much for your advice.
Comment 2.2: A relationship between platelet activation and VTE is still not clear.
Response: Thank you very much for your question. In the Pathophysiology of VTE in COVID‑19 (2.1. Platelet activation) section, we further discuss the factors of platelet activation in COVID-19, including the role of various pro-inflammatory cytokines and platelet activation caused by expression of cytokines following systemic inflammation and endothelial dysfunction. (lines 155-158; lines 161-165) Thank you very much for your advice.
Comment 2.3: Anticoagulant therapy was reported to be useful in severe cases but to be not useful in mild cases
Response: As for the comparison of anticoagulant therapy in severe and mild cases, we further provided literature evidence, indicating that anticoagulant therapy has a greater advantage in severe cases. (lines 269-274) Thank you very much for your question.
Comment 2.4: When did VTE occur in patients with COVID-19?
Response: Thank you very much for your question. We further discussed that the occurrence of VTE is more frequent in patients with severe COVID-19, and supplement the characteristics of VTE patients in the section of Introduction. (lines 75-79)
Comment 2.5: Which vessels were frequently obstructed by thrombosis?
Response: Thank you very much for your question. Through literature search, we further discussed the location of thrombus occurrence in pulmonary vessels, which is reflected in autopsy reports of COVID-19 patients. (lines 60-63)
Comment 2.6: Illustration for the mechanism for onset of VTE in patients with COVID-19 may be helpful.
Response: Thank you very much for your valuable suggestions. We have included the graphical abstract in the revised draft. Thanks again for your suggestions.
Comment 2.7: Relationship mortality and VTE may be important
Response: Thank you very much for your suggestions. As for the relationship mortality and VTE, studies have shown that COVID-19 patients who develop VTE are at higher risk of death, which we add to the discussion in the section of Introduction of the revised manuscript. (lines 70-73)
Comment 2.8: Risk factors for VTE in patients with COVID-19 may be helpful.
Response: Thank you very much for your suggestions. As for risk factors for VTE, we further enriched relevant factors in the section of Introduction of the revised manuscript, such as gender, obesity and other factors. (lines 75-77) Thanks again for your suggestions.
Comment 2.9: "Take home message" may be required.
Response: Thank you very much for your suggestion. We have modified the exposition of the conclusion in the section of CONCLUSION of the revised manuscript. Thank you again. (lines 338-353)
We tried our best to improve the manuscript. These changes will not influence the content and framework of the paper. We appreciate for Editors/Reviewers’ warm work earnestly, and hope that the correction will meet with approval.
Once again, thank you very much for your comments and suggestions.
Round 2
Reviewer 2 Report
This manuscript has been sufficiently improved.
I have no further comments.
This manuscript is a resubmission of an earlier submission. The following is a list of the peer review reports and author responses from that submission.